# Bark and Ambrosia Beetle Communities in Hazelnut Orchards of Italy and Georgia

**DOI:** 10.3390/insects16111148

**Published:** 2025-11-09

**Authors:** Eleonora Cresta, Nicolò Di Sora, Roberto Masturzi, Luca Rossini, Matteo Maspero, Marika Kodua, Tommaso De Gregorio, Miloš Petrović, Stefano Speranza, Mario Contarini

**Affiliations:** 1Dipartimento di Scienze Agrarie e Forestali, Università degli Studi della Tuscia, Via San Camillo de Lellis snc, 01100 Viterbo, Italy; eleonora.cresta@unitus.it (E.C.); roberto.masturzi@unitus.it (R.M.); speranza@unitus.it (S.S.); contarini@unitus.it (M.C.); 2National Biodiversity Future Center (NBFC), Piazza Marina 61, 90133 Palermo, Italy; 3Service d’Automatique et d’Analyse des Systèmes, Université Libre de Bruxelles, Av. F.D. Roosvelt 50, CP 165/55, 1050 Brussels, Belgium; luca.rossini@ulb.be; 4Hazelnut Company Division, Ferrero Group, 16 Rue de Trèves, L-2633 Senningerberg, Luxembourg; matteo.maspero@ferrero.com (M.M.); tommaso.degregorio@ferrero.com (T.D.G.); 5Hazelnut Company AgriGeorgia—111, Saint Nino Str. Dis., Chitatskari 2139, Georgia; marika.kodua@agrigeorgia.ge; 6Department for Environmental and Plant Protection, Faculty of Agriculture, University of Novi Sad, Trg Dositeja Obradovica 8, 21000 Novi Sad, Serbia; milos.petrovic@polj.edu.rs; 7Centro de Estudios Parasitológicos y de Vectores (CEPAVE, CONICET-UNLP), Blvd. 120 1900, La Plata B1900, Argentina

**Keywords:** Scolytinae, hazelnut orchards, pest management, trap performance

## Abstract

**Simple Summary:**

Hazelnut is a significant crop in many areas worldwide, but it is increasingly threatened by bark and ambrosia beetles. These beetles can damage trees directly and may also carry harmful fungi, making them a growing concern for farmers, especially under climate change conditions. Although several studies have investigated beetle species associated with hazelnut orchards, information on the overall beetle community and how different farming practices affect their populations is still limited. Our study explored beetle communities in hazelnut orchards located in Italy and Georgia, two regions with high production levels. We compared orchards with three types of management: integrated pest control, organic, and natural state. We also tested three different trap designs to see which captured the highest numbers of beetles. We found five species in total, with one species being the most common in both countries. Beetle numbers were lowest in organic orchards and highest in the other two systems. Traps varied in performance, with a simple handmade design working best. These results provide useful guidance to improve sustainable monitoring and protection of hazelnut crops.

**Abstract:**

Bark and ambrosia beetles (Coleoptera: Curculionidae: Scolytinae) are major pests of agricultural and forest systems and include some species which attack hazelnut (*Corylus avellana* L.) as well. Although hazelnut cultivation is increasingly threatened by insect and mite pests, yet little is known about how management practices can reduce infestations. This study investigated bark and ambrosia beetles in hazelnut orchards from two highly productive areas: Viterbo district (central Italy) and Samegrelo-Zemo Svaneti region (Republic of Georgia). Three phytosanitary management strategies were compared across 2023–2024: Integrated Pest Management, organic farming, and renaturalised orchards. Monitoring also evaluated three ethanol-baited trap designs: three designs in Italy (handmade red funnel [BR] and bottle [TT] and commercial multi-funnel [MF]) and only TT in Georgia. *Anisandrus dispar*, *Hypothenemus eruditus*, *Xyleborinus saxesenii*, and *Xylosandrus germanus* were detected in both countries, while *Xyleborus monographus* occurred only in Italy. *Xyleborinus saxesenii* was the dominant species in both regions. Beetle populations differed among management strategies, being lowest in organic orchards and highest in integrated and renaturalised systems. Trap performance also varied, with BR consistently outperforming commercial MF. These findings highlight the influence of management and monitoring tools on scolytine populations, offering practical insights for sustainable hazelnut production.

## 1. Introduction

Bark and ambrosia beetles (Coleoptera: Curculionidae: Scolytinae) represent a large and diverse group of wood-boring insects which often cause serious damage to crops and forests [1,2,3,4,5]. Although they usually attack stressed, dying, or dead trees [6,7], some species are significant primary pests that can weaken or kill healthy trees as well [8]. The structural weakening of the tree and the damage observed on infested plants is usually associated with two main factors. The first factor is the chewing activity carried out by adults, which dig oviposition galleries into the wood to protect eggs and juvenile stages. The second damaging factor is the pool of fungi commonly associated with bark and ambrosia beetles, which develop into the gallery and further weaken the plant [9].

In recent years, the global trade of wood products and climate change have promoted the spread of alien species [10] and the expansion of beetles beyond their native range [11], dramatically increasing the possibility of outbreak occurrence [8,12]. Scolytinae are among the most successful groups of invasive species [1,13,14], as they are supported by their small size, cryptic life cycle, and broad host range. Within Scolytinae, Xyleborini is the most species-rich tribe [15], which contains many exotic species that have successfully established in Europe [1]. This group of insects includes pest species, such as *Xylosandrus compactus* (Eichhoff) [16], *Xylosandrus crassiusculus* (Motschulsky) [17], *Xylosandrus germanus* (Blandford) [18], and *Euwallacea fornicatus* (Eichhoff) [19], currently considered transient and under official phytosanitary control [4,20].

Although bark and ambrosia beetles mainly infest forest environments, in recent years, they have become a relevant problem for agriculture as well. The short distance between natural forests and cultivated orchards, as well as exceptional increases in population due to extreme weather events, are two possible causes of infestations in many cultivations, including hazelnut (*Corylus avellana* L. 1753). Hazelnut is a high-revenue plant cultivated in several countries across temperate regions of both the northern and southern hemisphere [21]. The economic importance of hazelnut comes from its fruits, as it produces one of the most widely consumed and highly appreciated nuts worldwide [22]. In 2021, global hazelnut production reached approximately 1,077,000 tonnes, mostly concentrated in Türkiye (which accounted for the largest share of production), and followed by Italy, US, Azerbaijan, and Republic of Georgia [23].

The growing interest behind this cultivation is leading to a global increase in the production [24] outside its native range [21]. This phenomenon of expansion, however, is constantly exposing this plant to various damaging agents which can compromise production efforts. Among the biotic adversities, insect and mite pests cause the major impact on the yield [25], followed by fungal and bacterial pathogens [26,27].

Phytosanitary management in hazelnut orchards is essential to ensure sustainable, healthy, and profitable production [25]. Integrated Pest Management (IPM) and organic farming are among the most widely adopted approaches, both focusing on ecosystem-based methods which reduce the impact of pests and pathogens. As the use of traditional agrochemicals is banned, pest control in organic farming primarily relies on natural enemies, plant varieties with low susceptibility, and biotechnical methods. Conversely, IPM adopts a more flexible approach which allows chemical control strategies, when necessary, only based on threshold levels and environmental impact assessments [28]. These management strategies involve a range of preventive and monitoring actions aimed at the early detection of pests and diseases, either through a direct identification of the organisms involved or through signs and symptoms on the plant. Baited traps, which sometimes incorporate chromotropic cues, represent a well-established method for continuous surveillance of bark and ambrosia beetles, supporting timely and selective measures in line with integrated or organic management principles.

This study, carried out over the growing seasons 2023 and 2024, aimed to investigate three relevant aspects behind the incidence of ambrosia and bark beetle communities in hazelnut orchards. Notably, the first objective of this paper was the determination of species composition and population size across orchards under different phytosanitary management practices: IPM, organic farming, and renaturalised (plots abandoned over three decades ago). The second objective was to compare three different types of traps, in the same hazelnut orchards, to assess their effectiveness. The third objective was a cross-country comparison of the ambrosia and bark beetles’ community in Italy and Republic of Georgia, within the core of the hazelnut-productive areas. The first two objectives of the study involved orchards from Central Italy, while the third one involved orchards from Central Italy and Samegrelo-Zemo Svaneti region of the Republic of Georgia.

## 2. Materials and Methods

### 2.1. Study Sites

#### 2.1.1. Italian Sites

A set of representative orchards in the Viterbo hazelnut district (Central Italy) have been selected to compare three different phytosanitary management strategies: IPM, organic farming (ORG), and renaturalised (REN) systems (Table 1). IPM orchards were managed according to the official guidelines issued by the Lazio Region [29], which define the active ingredients allowed and regulate their application frequency to ensure sustainable pest control. For the sake of this study, three IPM orchards were selected and monitored throughout its overall duration. Organic hazelnut orchards were selected among farms certified by independent bodies under EU organic farming regulations, which guarantee the long-term use of practices that exclude traditional agrochemicals. Both IPM and ORG orchards were generally planted at a 5 × 5 m spacing. The growing season 2023 involved three sites (ORG_1, ORG_2, ORG_3), while in 2024, due to farm availability issues, ORG_2 was replaced with a different orchard, ORG_4. IPM and organic orchards ranged in size between 10,000 and 60,000 m^2^ and were uniformly composed of multi-stemmed hazelnut plants with an average age of approximately 30 years. The surrounding landscape is largely dominated by hazelnut orchards under various management regimes, alternated with vineyards, olive groves, and herbaceous crops. In two sites (IPM_2 and ORG_3), patches of natural forest formations, mainly composed of *Quercus cerris* L. and *Quercus pubescens* Willd., are also present.

Three renaturalised orchards (Figure 1A, Table 1), covering areas between 3000 and 20,000 m^2^ and consisting of 50-year-old hazelnut trees planted at a 4 × 4 m spacing, were selected within the Regional Park “Valle del Treja”. These orchards were abandoned in the early 1990s due to mechanization constraints and are currently undergoing natural ecological succession. The resulting habitats gradually evolved into structurally complex and floristically rich Mediterranean forest ecosystems, where naturalized *Corylus avellana* coexists with several tree species (e.g., *Quercus* spp., *Fraxinus ornus* L., *Ostrya carpinifolia* Scop., *Ulmus minor* Mill., *Acer* spp.), shrubs (e.g., *Rubus* spp., *Arbutus unedo* L., *Rhamnus* spp., *Ruscus aculeatus* L.), and herbaceous species (e.g., *Ranunculus* spp., *Pteridium aquilinum* (L.) Kuhn, *Cyclamen* spp., *Rubia peregrina* L.).

A summary of typical monthly minimum and maximum temperatures values for each location is reported in Appendix A.

#### 2.1.2. Georgian Sites

Monitoring activities in the Republic of Georgia were conducted in the Samegrelo-Zemo Svaneti region and involved six hazelnut orchards managed under IPM (Figure 1B, Table 1). The orchards were selected according to the following common characteristics: hazelnut plants were mature and in full production, belonging to local cultivars (predominantly Anakliuri), and were trained as multi-stemmed open vase trees, arranged in a regular 5 × 3 m planting layout.

A summary of typical monthly minimum and maximum temperature values for the main locations is reported in Appendix A.

### 2.2. Trap Types, Experimental Scheme and Sample Analysis

Monitoring activities in the Italian site involved three different trapping systems designed to assess bark and ambrosia beetle populations: two handmade models and a commercially available trap (Figure 2). This part of the study, carried out only in the Viterbo hazelnut district, involved a comparison of the effectiveness, in terms of catching potential, between the three models hereafter described.

The bottle trap (henceforth TT trap) (Figure 2A) is made of two transparent plastic bottles connected mouth-to-mouth through a Tornado Tube^®^ (Steve Spangler Science, Englewood, CO, USA). Three side openings (10 × 6 cm each) allowed insects to enter the trap from the upper bottle (1.5 L). The lower bottle (0.5 L) worked as a collection container, and it was periodically filled with 200 mL of a 3% aqueous dish detergent solution (henceforth soapy solution) to preserve the specimens caught. The structure is enclosed and covered with a saucer (25 cm diameter) that prevents rainwater from entering the container.A handmade trap (henceforth BR trap) (Figure 2B), partially inspired by the commercial BROCAP^®^ trap, was designed and manufactured. The BR trap was made of a 20 cm diameter red saucer attached to two red polycarbonate panels (20 × 15 cm) using cable ties. Each panel featured a central window of approximately 10 × 10 cm. The panels were placed orthogonally and connected to a 15 cm diameter red funnel, also secured with cable ties. The funnel was then connected to a Falcon tube, working as a collection container, through its perforated cap. Each Falcon tube was partially filled with the 3% soapy solution to ensure preservation of the samples.The multi-funnel trap (henceforth MF trap) (Figure 2C), provided by ChemTica Internacional (Santo Domingo, Costa Rica), was made of 12 vertically aligned interconnected black plastic funnels (20 cm diameter each). A removable collection cup, equipped with a mesh bottom to allow rainwater drainage, is securely screwed in place to collect and retain the captured specimens.

In the Italian sites, traps were deployed following a randomized block design, where each block included one trap of each type. Within each block, traps were spaced approximately 5 m apart, with a minimum distance of 80 m between blocks. Each block was positioned at least 20 m from the edge of the hazelnut orchard to avoid border effects (Figure 3A). Randomized blocks were nested within sites, and both sites and blocks were monitored in both years, maintaining the same spatial design. In 2023, five replicates per trap type were deployed, and three in 2024. Traps were deployed on 21 February 2023 and 20 March 2024, and removed on 9 November 2023 and 30 October 2024, respectively.

In the Georgian sites, monitoring involved three TT traps per site, spaced at least 50 m apart and located at a minimum distance of 20 m from the orchard edges (Figure 3B). Traps were deployed on 15 May 2023 and on 15 March 2024, and removed on 18 October 2023 and 13 October 2024, respectively.

All traps were hung at a height of 2 m above the ground, directly to a hazelnut tree branch and were baited with polyethylene diffusion bags containing 5 mL of 96% ethanol and a roll of dental-grade cotton wool. The bags with lures were hung in the upper bottle of the TT trap, in the central window of the BR trap, and in the second-to-last funnel of the MF trap, respectively. Traps were inspected every 15 days to replace the ethanol lures and to collect trapped insects, which were subsequently preserved in 70% ethanol until identification. After each inspection, the lures were completely refilled with new 5 mL of 96% ethanol. Ethanol mass loss was not systematically monitored across sites. The specimens collected were properly sealed and brought to the laboratory where they were identified and counted using a Leica EZ 4W stereo dissecting microscope (Leica Microsystems (Schweiz) AG, Heerbrugg, Switzerland) and specific dichotomous keys [30,31,32,33]. All specimens were identified by the authors, with the exception of *Hypothenemus* specimens, which were sent to and identified by Massimo Faccoli (Università degli Studi di Padova).

### 2.3. Data Analysis

Data analysis was carried out through the R software (R Development Core Team, version 2022) to assess the following three main research questions of this study: (i) the effect of phytosanitary management strategies on bark and ambrosia beetles communities, (ii) differences, in terms of efficacy, between the three traps described in Section 2.2, and (iii) differences, in terms of bark and ambrosia beetle communities, between Italian and Georgian hazelnut orchards. The datasets and the scripts to fully reproduce the results of this study are publicly available at (https://github.com/eleonoracresta/BeetleCommunitiesHazelnutOrchards (accessed on 25 September 2025)).

#### 2.3.1. The Effect of Phytosanitary Management Strategies on Bark and Ambrosia Beetle Communities and Populations

Data were first evaluated for overdispersion, which was detected and addressed by fitting a generalized linear mixed model with a negative binomial distribution, including random effects (GLMER; lme4 package, version 1.1.35.5). Residual diagnostics were evaluated using the DHARMa package (version 0.4.7), including tests for overdispersion, uniformity, and zero inflation. Confidence intervals (CIs—95%) for fixed-effect parameters were calculated on the log scale using Wald approximations. The first analysis considered the phytosanitary management strategy as independent variables and the number of individuals collected per trap and sampling date as the dependent variable. The model included random factors for sampling year and for blocks nested within sites, to account temporal and spatial dependencies in the data. Calculations were carried out through the glmer.nb() function within the lme4 package in R [34]. Pairwise comparisons were conducted using the emmeans() function within the R package emmeans [35], with *p*-values adjusted using the Benjamini–Hochberg correction (*α* = 0.05). Additional functions included pairs() from the R package multcompView [36], and cld() from the R package multcomp [37] to visualize statistically significant groupings.

A second set of models was applied at the species level, testing each beetle species separately across the three management types. Data were analyzed using a GLMER with a negative binomial distribution, after verifying the presence of overdispersion, following the same approach as for the previous analyses, with residual diagnostics and 95% CIs evaluated in the same manner. The number of individuals of each species collected per trap at each sampling date was taken as the dependent variable, the management strategy as the independent variable, and included random factors for sampling year and for blocks nested within sites.

The beetle communities under the three management types were analyzed using community-level approaches. Alpha diversity was calculated using the Shannon and Simpson indices, and differences among management types were tested using the Kruskal–Wallis test. Post hoc pairwise comparisons were performed using Dunn’s test with Benjamini–Hochberg adjustment. Beta diversity was assessed using Bray–Curtis dissimilarity, and compositional differences among management types were tested with PERMANOVA (vegan package, version 2.6.8). Community structure was visualized using non-metric multidimensional scaling (NMDS) based on Bray–Curtis distances; in the NMDS ordination, ellipses represent 95% confidence intervals for each management type. Finally, to identify species indicative of each management type, we applied the IndVal procedure of Dufrêne and Legendre [38] and its generalization for multiple and unequal groups, IndVal.g [39], at the species level. Only species with significant associations (*p* < 0.05) were considered as indicators of management type or combination of types.

#### 2.3.2. Differences Between the Three Trap Types (TT, BR, and MF)

Data were analyzed through a GLMER with a negative binomial distribution, after verifying the presence of overdispersion, in this case as well. Residual diagnostics were evaluated using the DHARMa package, including tests for overdispersion, uniformity, and zero inflation. Confidence intervals (CIs—95%) for fixed-effect parameters were calculated on the log scale using Wald approximations. The trap type was considered as the independent variable, and the number of individuals collected per trap at each sampling date was used as the dependent variable. The model included random effects for sampling year and for blocks nested within sites. Pairwise comparisons with Benjamini–Hochberg correction were applied in the same way as described in Section 2.3.1.

#### 2.3.3. Differences in Composition and Abundance of Bark and Ambrosia Beetle Species Between Hazelnut Orchards in Italy and Republic of Georgia

Prior to the country-level comparisons, separate GLMERs with a negative binomial distribution, after verifying the presence of overdispersion, were fitted to assess the abundance of bark and ambrosia beetle species within each country. Residual diagnostics were then evaluated using the DHARMa package, including tests for overdispersion, uniformity, and zero inflation. Confidence intervals (CIs—95%) for fixed-effect parameters were calculated on the log scale using Wald approximations. In these models, the species identity was considered as the independent variable, the number of individuals collected per trap at each sampling date was used as the dependent variable, and random effects included sampling year and sites. Pairwise comparisons with Benjamini–Hochberg correction were applied in the same way as described in Section 2.3.1.

For the country-level comparisons, data were analyzed using a GLMER with a negative binomial distribution, after verifying the presence of overdispersion, following the same approach as for the previous analyses, with residual diagnostics and 95% CIs evaluated in the same manner. In addition, to account for differences in trapping duration among sites and years, the total number of trap-days was included as an offset term (offset(log()). The effect of country was tested using the number of individuals collected per trap at each sampling date was used as the dependent variable, with country as the independent variable. The model included random effects for sampling years and for sites. Pairwise comparisons with Benjamini–Hochberg correction was applied in the same way as described in Section 2.3.1.

A second set of models was applied at the species level, testing each beetle species separately between the two countries. All procedures described above, including the use of negative binomial GLMERs, overdispersion checks, residual diagnostics, 95% CIs calculation, and inclusion of trap-days as an offset, were applied in the same way to the species-level models. In this case, the number of individuals of each species collected per trap at each sampling date was used as the dependent variable, the country as the independent variable, and included the same random effect structure. Pairwise comparisons with Benjamini–Hochberg correction were applied in the same way as described in Section 2.3.1.

## 3. Results

### 3.1. Bark and Ambrosia Beetle Species in Hazelnut Orchards of Italy and Republic of Georgia

Five species of bark and ambrosia beetles were caught in the Italian sites, for overall 29,063 individuals; in the Georgian sites, four species were caught, for overall 7679 individuals collected. The species *Anisandrus dispar* (Fabricius), *Hypothenemus eruditus* Westwood, *Xyleborinus saxesenii* (Ratzeburg), and *Xylosandrus germanus* were collected in both countries, while *Xyleborus monographus* (Fabricius) was caught exclusively in the Italian sites. Residual diagnostics for the Italian and Georgian species models indicated no significant overdispersion in either model (Italy: dispersion = 0.28, *p* = 0.23; Georgia: dispersion = 2.07, *p* = 0.11), but deviations from a uniform distribution of residuals (Kolmogorov–Smirnov test; Italy: D = 0.02, *p* < 0.001; Georgia: D = 0.05, *p* < 0.01). No substantial zero inflation was observed in neither model (Italy: ratio = 0.99, *p* = 0.72; Georgia: ratio = 0.95, *p* = 0.23), and the overall model fit was considered adequate.

In Italy, the abundances of bark and ambrosia beetle species were statistically different (GLMER; *A. dispar*: Z = −4.76, *p* < 0.001; *H. eruditus*: Z = −14.70, *p* < 0.001; *X. germanus*: Z = 14.43, *p* < 0.001; *X. monographus*: Z = −20.21, *p* < 0.001; *X. saxesenii*: Z = 44.00, *p* < 0.001) (Appendix B, Table A2 and Table A3). The most and the least abundant species were *X. saxesenii* and *X. monographus* (Figure 4A), which have reported the highest and the lowest mean catches per trap, respectively. Conversely, in the Republic of Georgia statistical differences were observed among all the species (GLMER; *A. dispar*: Z = −8.60, *p* < 0.001; *H. eruditus*: Z = 16.36, *p* < 0.001; *X. germanus*: Z = 16.37, *p* < 0.001; *X. saxesenii*: Z = 19.21, *p* < 0.001) (Appendix B, Table A4), except between *X. germanus* and *H. eruditus* (*X. germanus* vs. *H. eruditus*: *p* = 0.67) (Appendix B, Table A5). *Xyleborinus saxesenii* and *A. dispar* were the most and the least abundant species, respectively (Figure 4B).

### 3.2. Influence of the Different Phytosanitary Management Strategies

Residual diagnostics indicated no overdispersion (dispersion = 0.95, *p* = 0.77), uniform distribution of residuals (Kolmogorov–Smirnov test, D = 0.01, *p* = 0.15), and negligible zero inflation (ratio = 0.99, *p* = 0.60), supporting an adequate fit of the negative binomial GLMER. The phytosanitary management strategies significantly affected the occurrence of bark and ambrosia beetles (GLMER; ORG: Z = −1.39, *p* = 0.17; IPM: Z = 2.39, *p* < 0.05; REN: Z = 2.12, *p* < 0.05), with IPM and REN showing higher values than ORG (Appendix C, Table A6), as also indicated by Post hoc pairwise comparisons with Benjamini–Hochberg correction, which revealed marginal differences between ORG and IPM (*p* = 0.05) and between ORG and REN (*p* = 0.05). No significant difference was observed between IPM and REN management (*p* = 0.77) (Figure 5) (Appendix C, Table A7).

The influence of phytosanitary management was further assessed by grouping the dataset per species (Appendix D, Figure A1) according to the results showed in Section 3.1, with the exception of *X. monographus*, which reported extremely low capture rates. For the species-specific models, residual diagnostics showed no evidence of overdispersion (dispersion range: 0.001–0.96, all *p* > 0.73) and no evidence of zero inflation (ratio range: 0.94–0.998, all *p* > 0.46). Residuals were generally uniform (Kolmogorov–Smirnov D = 0.012–0.023, *p* ≥ 0.05), with only a minor deviation detected in *X. saxesenii*, supporting an overall adequate fit of the negative binomial GLMERs for all species.

According to species-specific models, management type had a significant effect on beetle abundance for *X. saxesenii* and *X. germanus*, while no significant effect was detected for *A. dispar* or *H. eruditus* (GLMER; *X. saxesenii*: ORG: Z = 3.06, *p* < 0.01; IPM: Z = 2.75, *p* < 0.01; REN: Z = 1.21, *p* = 0.23; *A. dispar*: ORG: Z = −4.54, *p* < 0.001; IPM: Z = 1.05, *p* = 0.29; REN: Z = 2.24, *p* < 0.05; *X. germanus*: ORG: Z = −3.38, *p* < 0.001; IPM: Z = −0.98, *p* = 0.33; REN: Z = 5.20, *p* < 0.001; *H. eruditus*: ORG: Z = −2.47, *p* < 0.05; IPM: Z = −1.42, *p* = 0.16; REN: Z = 0.35, *p* = 0.73) (Appendix C, Table A8, Table A9, Table A10 and Table A11). Pairwise comparisons with BH adjustment showed that *X. saxesenii* populations were significantly higher in IPM than ORG orchards (ORG vs. IPM: *p* < 0.05) (Figure 6A) (Appendix C, Table A12), *X. germanus* abundance was significantly higher in REN orchards compared to ORG and IPM (BIO vs. RIN: *p* < 0.001; INT vs. RIN: *p* < 0.001) (Figure 6C) (Appendix C, Table A13), while no significant differences were observed among management types for A. dispar (BIO vs. INT: *p* = 0.29; BIO vs. RIN: *p* = 0.08; INT vs. RIN: *p* = 0.29) (Figure 6B) (Appendix C, Table A14) or *H*. *eruditus* (BIO vs. INT: *p* = 0.233; BIO vs. RIN: *p* = 0.730; INT vs. RIN: *p* = 0.233) (Figure 6D) (Appendix C, Table A15).

Alpha diversity differed significantly among management types, with both the Shannon and Simpson indices showing the same pattern (Kruskal–Wallis test: χ^2^ = 7.20, *df* = 2, *p* < 0.05). Post hoc Dunn’s tests with BH adjustment revealed that diversity was significantly higher in REN orchards compared to IPM (*p* < 0.05), while no significant differences were observed between ORG and IPM (*p* = 0.18) or between ORG and REN (*p* = 0.27) for either index (Figure 7) (Appendix E, Table A16 and Table A17). Beta diversity also differed significantly among management types, as assessed by PERMANOVA based on Bray–Curtis dissimilarities (adonis2: df = 2, F = 3.33, R^2^ = 0.53, *p* < 0.05, 999 permutations) (Appendix E, Table A20). Community composition was visualized using non-metric multidimensional scaling (NMDS) (Figure 8). In the NMDS ordination, ellipses represent 95% confidence intervals for each management type, showing distinct clustering of sites according to management. These results indicate that both the diversity and the composition of beetle communities varied according to management type. Finally, indicator species analysis using the IndVal.g method (999 permutations) did not identify any species significantly associated with a specific management type or combination of types (*p* ≥ 0.05).

### 3.3. Different Types of Traps

Residual diagnostics indicated no overdispersion (dispersion = 0.90, *p* = 0.80), minor deviations from a uniform distribution of residuals (Kolmogorov–Smirnov test, D = 0.01, *p* < 0.05), and negligible zero inflation (ratio = 0.99, *p* = 0.49), supporting an adequate fit of the negative binomial GLMER.

Trap type significantly affected the total number of ambrosia and bark beetles caught (GLMER; BR: Z = 1.03, *p* = 0.30; MF: Z = −5.24, *p* < 0.001; TT: Z = −0.74, *p* = 0.46) (Appendix F, Table A19). Post hoc comparisons showed that BR and TT traps caught significantly more individuals than MF traps (BR vs. MF: *p* < 0.001; TT vs. MF: *p* < 0.001), while BR and TT did not differ from each other (BR vs. TT: *p* = 0.46) (Figure 9) (Appendix F, Table A20).

### 3.4. Comparison Between Italy and Republic of Georgia

Residual diagnostics indicated no significant overdispersion (dispersion = 1.59, *p* = 0.21) and no evidence of zero inflation (ratio = 0.99, *p* = 0.75). A minor deviation from residual uniformity was observed (Kolmogorov–Smirnov test, D = 0.023, *p* < 0.05), but overall, the model fit was considered adequate.

The cross-country comparison between Italy and Republic of Georgia involved only ambrosia and bark beetle species detected in both countries. Moreover, we have considered only Italian sites under IPM phytosanitary management and TT traps, for a peer comparison. The total number of ambrosia and bark beetles caught was significantly different between the two countries (GLMER; Republic of Georgia: Z = −18.59, *p* < 0.001; Italy: Z = −4.21, *p* < 0.001) (Appendix G, Table A21), with higher mean captures observed in Republic of Georgia than Italy (*p* < 0.001) (Appendix G, Table A22) (Figure 10).

For the species-specific models, residual diagnostics indicated that all negative binomial GLMERs adequately fitted the data. The model for *X. saxesenii* showed moderate overdispersion (dispersion = 2.35, *p* < 0.05) but no evidence of zero inflation (ratio = 0.90, *p* = 0.26) and fairly uniform residuals (Kolmogorov–Smirnov test, D = 0.04, *p* = 0.14). Models for *X. germanus*, *A. dispar*, and *H. eruditus* showed no overdispersion, no zero inflation, and uniform residuals (all *p* > 0.05), supporting an overall adequate fit for assessing country-related differences.

A more detailed comparison of species abundance between the two countries revealed statistically significant differences in all cases, except for *X. saxesenii*, whose abundance tended to be higher in Georgia than in Italy, although the difference was only marginally non-significant (GLMER; Georgia: Z = −15.73, *p* = < 0.001; Italy: Z = 1.89, *p* = 0.06; Georgia vs. Italy: *p* = 0.06) (Figure 11A) (Appendix G, Table A23 and Table A24).

*X. germanus* (GLMER; Georgia: Z = −7.66, *p* < 0.001; Italy: Z = −4.48, *p* < 0.001; Georgia vs. Italy: *p* < 0.001) (Figure 11C) (Appendix G, Table A27 and Table A28), and *H. eruditus* (GLMER; Georgia: Z = −7.81, *p* < 0.001; Italy: Z = −5.73, *p* < 0.001; Georgia vs. Italy: *p* < 0.001) (Figure 11D) (Appendix G, Table A29 and Table A30) showed higher mean abundances in Georgia, while *A. dispar* (GLMER; Georgia: Z = −18.68, *p* < 0.001; Italy: Z = 1.95, *p* = 0.05; Georgia vs. Italy: *p* = 0.05) (Figure 11B) (Appendix G, Table A25 and Table A26) was more abundant in Italy.

## 4. Discussion

This study aimed to assess the most recurrent bark and ambrosia beetle species in hazelnut orchards of two high-productive countries (Italy and the Republic of Georgia), the effect of the phytosanitary management strategy on the species abundance, and the efficacy of three different trap designs. Results showed differences in terms of composition and abundance of bark and ambrosia beetle communities between Italy and the Republic of Georgia, differences in terms of trap efficacy, and different effects of phytosanitary management strategies on population abundances. This pool of results provides relevant information on bioecological aspects related to bark and ambrosia beetles and helpful indications to plan monitoring activities.

Five and four species were detected in Italy and in the Republic of Georgia, respectively; *X. monographus* was exclusively found within Italian orchards, while the other four species were detected in both countries. *Xyleborus monographus* is a recurrent species in Mediterranean habitats primarily associated with *Quercus* spp. [40] and other broadleaf trees typical of these woodlands, likely explaining its absence in Georgian orchards. The consistently high abundance of *X. saxesenii* across both countries could be justified by its ecological flexibility and wide host range [41,42] which is known to support its adaptation to different environments. *Xylosandrus germanus* is a highly generalist species native to Asia which showed, in many cases, its capability of establishing in a variety of habitats, including the European continent [43]. Notably, according to Alonso-Zarazaga et al. [44], only *X. germanus* has been formally reported in Georgia. However, other species have been documented in the literature [45,46], confirming their presence in the country. Conversely, the lower abundance of *H. eruditus* and *A. dispar* might suggest more specific habitat requirements or lower attraction to the traps systems considered. The scenarios observed in our study are in line with similar studies conducted by Tuncer et al. [32] in hazelnut orchards in Türkiye.

Although similar sampling protocols, the overall number of specimens caught were significantly higher in the Republic of Georgia than in Italy. This gap may be the effect of different combinations of environmental factors, such as greater forest cover, milder microclimates, or differences in orchard structure and tree health. Interestingly, *A. dispar* is more abundant in Italian orchards, probably because of its strong association with broadleaf trees commonly found in Mediterranean environments. Its reduced presence in Georgian sites may be due to a lower host plant availability or to a different forest composition. *Anisandrus dispar* is recognized as an injurious pest of fruit and hazelnut trees [47,48], typically infesting trees weakened by biotic and/or abiotic factors [49]. The higher occurrence of *A. dispar* in Italian orchards, combined with its known capability to colonize thinner branches typical of hazelnut trees [49], suggests a greater potential impact on hazelnut production compared to the risk posed by the more abundant *X. saxesenii*, which is rarely considered a significant pest [48].

Phytosanitary management influenced beetle communities, primarily through differences between IPM and renaturalised (REN) orchards. Alpha diversity, measured by both Shannon and Simpson indices, was significantly higher in REN orchards compared to IPM, while organic (ORG) orchards showed intermediate values. Similarity, beta diversity varied among management types, indicating that both the richness and composition of beetle assemblages were affected by management. NMDS ordination further revealed clear clustering of sites according to management type. Despite these patterns, indicator species analysis did not detect any species uniquely associated with a specific management type or combination of types, suggesting that while management shapes overall community structure, individual species may be widely distributed, potentially due to their opportunistic ecology or broad host range.

IPM and renaturalised orchards generally showed higher overall beetle populations than organic orchards. This pattern seems to be associated with differences in habitat complexity and ecological stability among management types. In particular, sites showing a higher presence of stressed host plants, potentially related to management practices such as mechanization, non-professional pruning, and inadequate irrigation management in IPM orchards, as well as greater structural diversity, typically found in more naturalized environments, such as REN sites, might offer conditions known to enhance bark and ambrosia beetles attraction and proliferation. In integrated orchards, the moderate but consistent use of plant protection products aimed at controlling primary pests, combined with frequent and intense disturbance regimes, might be associated with reduced populations of natural enemies, indirectly influencing beetle abundance [50,51]. This factor, combined with potential plant stress induced by intensive cultural practices, may favour opportunistic species such as *X. saxesenii* [52]. Conversely, organic orchards, which refrain from chemical inputs and prioritize ecological balance, may sustain more stable and diverse communities of natural antagonists, effectively limiting beetle populations [53]. In renaturalised orchards, the ongoing transition from cultivated hazelnut groves to mixed woodland likely results in a higher diversity of woody plant species. This increased structural complexity may offer a broader range of ecological niches [54], thereby supporting more diverse and abundant Scolytinae populations. Additionally, successional vegetation dynamics may be altering the phytosanitary status of hazelnut trees, which now experience competition from other plant species. These stressed or senescent hazelnut trees may serve as functional hosts, promoting the development of ambrosia beetle populations.

Although only a long-term study can provide a comprehensive understanding of the impact of plant protection practices on arthropod communities inhabiting hazelnut orchards, useful insights are offered by the study of Contarini et al. [55]. The authors reported that, in the same integrated, organic and renaturalised orchards examined in the present work, infestations by the big bud mite, *Phytoptus avellanae*, were significantly lower in renaturalised systems. This difference seems to be driven by the contrasting effects of plant protection strategies on the communities of the mite’s natural enemies, which tend to be favoured in management regimes with reduced chemical input.

Trapping-based monitoring approaches have been extensively investigated in several studies [18,56] as their effectiveness is highly context-dependent. It is well established that multiple variables can influence their performance, making it essential to continuously update these methodologies, particularly in agriculture systems, to adapt to the evolving conditions that may arise in the application environment [57,58]. For instance, a closed canopy resulting from a dense orchard planting pattern or from non-regular and non-intensive pruning practices may create favourable humid conditions for the development of bark and ambrosia beetle populations [59]. Similarly, the proximity of hazelnut orchards to mixed forested areas may contribute to a greater incidence of xylomycetophagous taxa. In this context, updating and adapting monitoring tools becomes particularly relevant for hazelnut agroecosystems.

Additional environmental variables, such as proximity to natural forests, canopy closure and availability of deadwood, can significantly influence bark and ambrosia beetle communities. These factors were not quantified in the present study and could not be included as covariates in the models. Future studies should incorporate such variables to further refine the understanding of management effects on beetle populations. Moreover, it is worth noting that ethanol evaporation rates may have differed among sites due to microclimatic variation (e.g., temperature, humidity, and exposure). Since traps were baited with a fixed 5 mL of 96% ethanol renewed approximately every 15 days, differences in evaporation might have affected the effective release rate of the attractant, potentially influencing capture efficiency. While this factor is unlikely to alter the overall interpretation of management-related patterns, it should be acknowledged as a potential limitation influencing the comparability of capture rates across environments.

This study provides an updated evaluation that reflects the current species assemblage and monitoring needs in hazelnut orchards. The results of this study show that trap type had a significant effect on the number of ambrosia and bark beetles caught. In particular, BR and TT traps outperformed MF traps, catching significantly higher numbers of individuals, with no significant different between BR or TT traps. These results demonstrate that well-designed handmade traps can be as effective as commercial traps for monitoring ambrosia and bark beetles, and in some cases may even outperform them, offering a cost-effective and flexible alternative for pest surveillance programmes.

In this case, given the more consistent performance of BR and TT traps compared to MF, the observed differences cannot be attributed solely to trap colour, as these traps also differ in structure and design. These findings suggest that both trap design and colour strongly affect trapping efficiency. In fact, the observed differences are the result of the combined effects of visual attraction and trap architecture. Visual attraction is mainly linked to colour, which may determine the initial detection and approach distance of flying adults, whereas the shape of the trap and the configuration of its entrance can influence the probability that insects will enter and remain inside. In addition, trap geometry may also affect air circulation and the dispersion of the lure, further contributing to differences in trap performance. In both BR and TT traps, the lure bag was placed in a central position, favouring both odour diffusion and insect guidance toward the entrance, thus enhancing capture success. Furthermore, in both types of traps, the collection container held a soapy solution, which may have contributed to retaining the trapped insects more effectively. Further studies set up on crossed experimental designs would be needed to disentangle the effects of colour and structure.

From a practical standpoint, both BR and TT traps represent suitable options for growers and technicians involved in ambrosia and bark beetle monitoring. Both traps have significantly lower construction costs than commercial models, such as MF, and costs can be almost negligible when the TT trap is assembled from recycled plastic bottles. These handmade traps are easy to assemble and to deploy in the field. BR traps proved to be more durable, as they were used in the field for several years, whereas TT traps needed to be replaced after one or two years.

To the best of our knowledge, this study provides the first systematic comparison of trap types in hazelnut orchards under current conditions, influenced by the introduction of invasive organisms such as *Xylosandrus* species and the effects of climate change. Although Speranza et al. [60] previously conducted a comparative trial in hazelnut orchards of central Italy, that work was carried out almost two decades ago, when invasive bark and ambrosia beetles such as *Xylosandrus germanus* were not yet established in the area. During the monitoring activity, no specimens of the invasive species *X. compactus* or *X. crassiusculus* were collected, despite the widespread distribution in the Lazio Region [3] and their presence few kilometres apart from the hazelnut orchards under investigation (Cresta pers. obs.). Although hazelnut has not been reported as a host plant [61], the highly polyphagous nature of these species suggests that in the future they could potentially exploit this extensively cultivated crop.

Previous studies have also highlighted the effectiveness of red sticky traps in Turkish hazelnut orchards [62,63]. However, the practical application of sticky traps is limited because identification of the specimens is usually carried out directly in the field, which may lead to misidentification [32]. Although replacing traps after each inspection could mitigate this issue, it would considerably increase both costs and human efforts. In contrast, the traps evaluated in the present study allow the collected insects to be easily removed and preserved, enabling subsequent laboratory identification with greater accuracy. The results obtained with the BR trap seems to support a role of red colour in attracting bark and ambrosia beetles, which could enhance the effect of the chemical lure and overall trap performance. Interestingly, similar colour-related attraction patterns have been reported for other Scolytinae species, such as *Hypothenemus hampei* (Ferrari), which shows a strong preference for red traps compared to other colours [64,65]. This suggests that the positive response to red may represent a conserved visual cue within several ambrosia beetle taxa, potentially supporting the reliability of red traps for monitoring purposes across different host systems.

## 5. Conclusions

In conclusion, this study provides an updated assessment of bark and ambrosia beetle communities in hazelnut orchards of central Italy and Georgia, highlighting the importance of phytosanitary management, species ecology, and local environmental conditions in shaping beetle abundance. Integrated and renaturalised orchards generally supported higher beetle populations, likely due to host stress and increased structural complexity, whereas organic orchards sustained more stable communities of natural enemies, potentially limiting pest populations. Moreover, the use of handmade traps proved effective for bark and ambrosia beetle monitoring, providing valuable data on beetle ecology. These findings offer significant guidance for refining pest monitoring and management practices in hazelnut agroecosystems.

## Figures and Tables

**Figure 1 insects-16-01148-f001:**
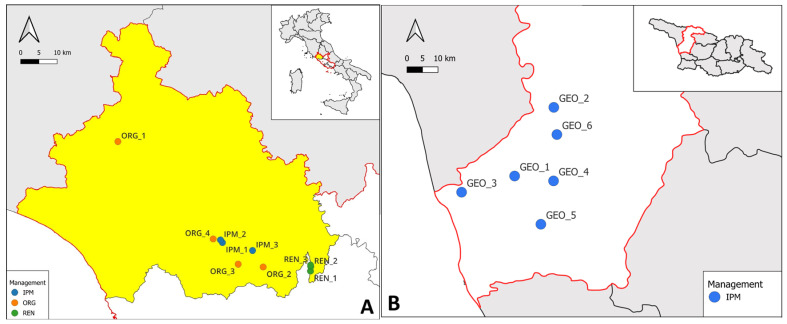
(**A**) Experimental orchards selected in the Viterbo province (in yellow), within the Lazio region (red boundaries), Central Italy. Orchards are managed under different phytosanitary strategies: Integrated Pest Management (IPM, blue), organic farming (ORG, orange), and renaturalised areas (REN, green); (**B**) Experimental orchards in the Samegrelo-Zemo Svaneti region (red boundaries), in western Republic of Georgia. All orchards are managed under Integrated Pest Management (IPM, blue). Geographical coordinates are provided in Table 1.

**Figure 2 insects-16-01148-f002:**
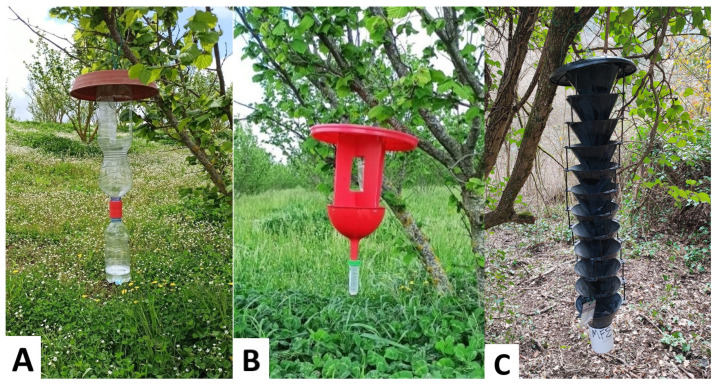
Types of traps used in this study: (**A**) TT trap (handmade trap), made of two transparent plastic bottles connected through a Tornado Tube^®^; (**B**) BR trap (handmade trap), consisting of red polycarbonate panels, a funnel, and a Falcon tube as collection container; (**C**) MF trap (commercial trap), provided by ChemTica Internacional, composed of 12 vertically aligned interconnected funnels with a removable collection cup.

**Figure 3 insects-16-01148-f003:**
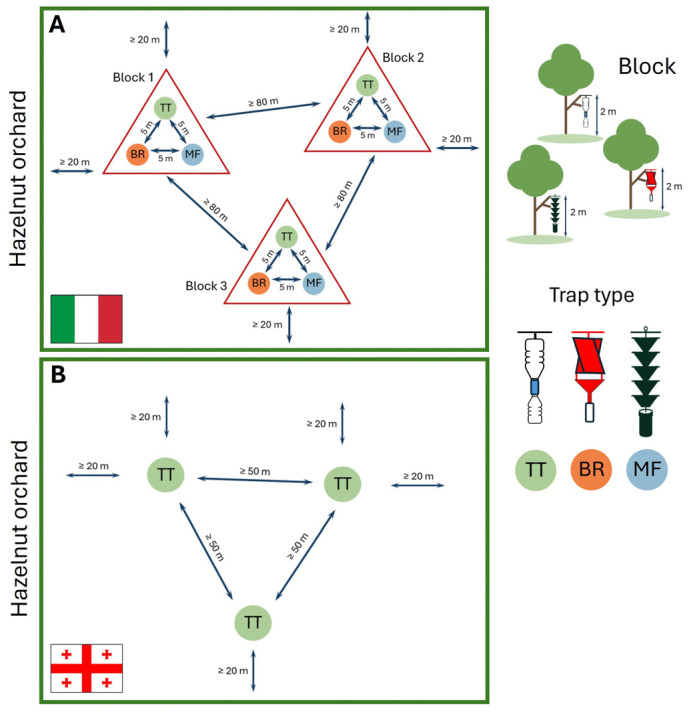
Trap configuration within the hazelnut orchards in Italy (**A**) and Republic of Georgia (**B**). In Italian sites, each block consists of three different trap types (TT, BR, MF) arranged in a 5 m sides triangle. Blocks are randomly distributed within the orchard, respecting a minimum distance of 80 m apart and 20 m from the edge of the orchard. In the Georgian sites, three TT traps per site were arranged with a minimum distance of 50 m between traps and 20 m from the orchard edge.

**Figure 4 insects-16-01148-f004:**
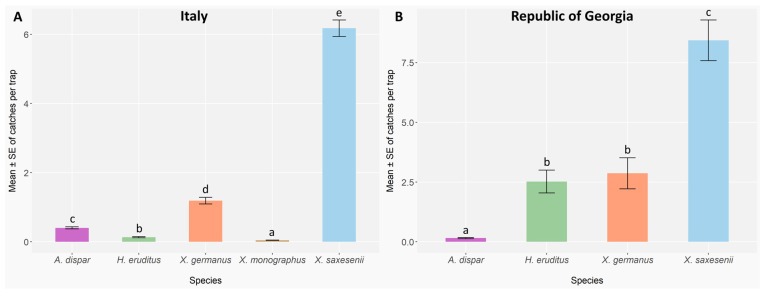
Mean abundance ± standard error (error bars) of ambrosia and bark beetle species caught per trap across sampling sites in Italy (**A**) and Georgia (**B**), over the two years of observations 2023 (Italy: n = 3 traps per block, 5 blocks per site, 9 sites, and 17 two-week checks; Georgia: n = 3 traps per site, 5 sites, and 11 two-week checks) and 2024 (Italy: n = 3 traps per block, 3 blocks per site, 9 sites, and 17 biweekly checks; Georgia: 3 traps per site, 4 sites, and 15 two-week checks). Different letters indicate significant differences between species after the Benjamini–Hochberg (BH) Post hoc test (*p* < 0.05).

**Figure 5 insects-16-01148-f005:**
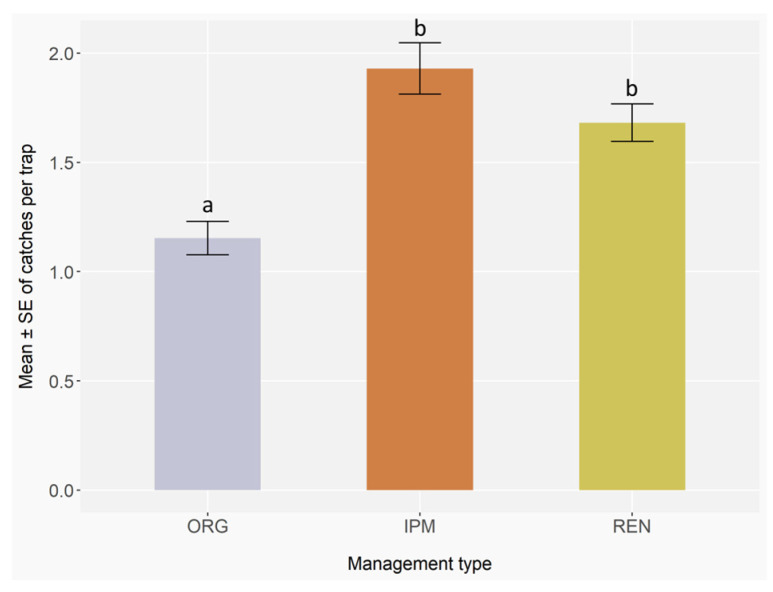
Mean number ± standard error (error bars) of ambrosia and bark beetles caught in orchards having different phytosanitary management strategies. ORG, IPM, and REN stands for organic farming, Integrated Pest Management, and renaturalised areas. Monitoring was conducted with 3 traps per block, 5 blocks per site, 9 sites, and 17 two-week checks in 2023, and 3 traps per block, 3 blocks per site, 9 sites, and 17 checks in 2024. Different letters indicate statistically significant differences between management types based on GLMER followed by the Benjamini–Hochberg Post hoc test (*p* < 0.05).

**Figure 6 insects-16-01148-f006:**
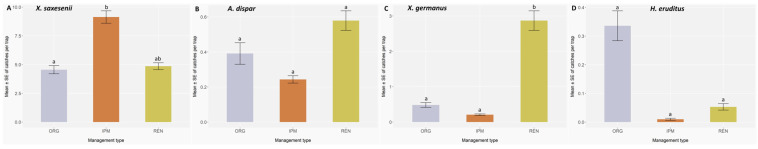
Mean number ± standard error (error bars) for each ambrosia or bark beetle species captured under different phytosanitary management regimes: (**A**) *Xyleborinus saxesenii*; (**B**) *Anisandrus dispar*; (**C**) *Xylosandrus germanus*; (**D**) *Hypothenemus eruditus*. Different letters indicate statistically significant differences between management types based on the Benjamini–Hochberg test (*p* < 0.05).

**Figure 7 insects-16-01148-f007:**
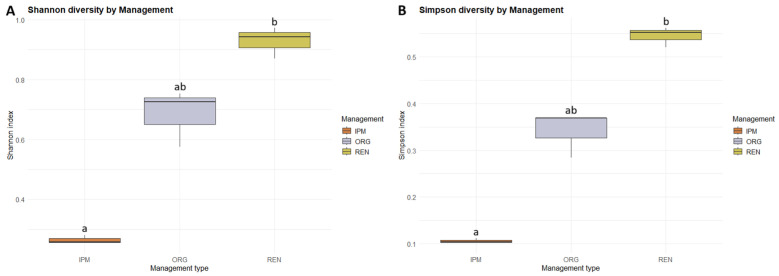
Boxplot of Shannon (**A**) and Simpson diversity (**B**) of ambrosia and bark beetle communities across three management types (ORG, IPM, REN). Boxes represent the interquartile range (IQR), horizontal lines indicate the median, and whiskers extend to 1.5 × IQR. Different letters indicate statistically significant differences between management types, based onKruskal–Wallis and Dunn’s Post hoc tests with Benjamini–Hochberg correction (*p* < 0.05).

**Figure 8 insects-16-01148-f008:**
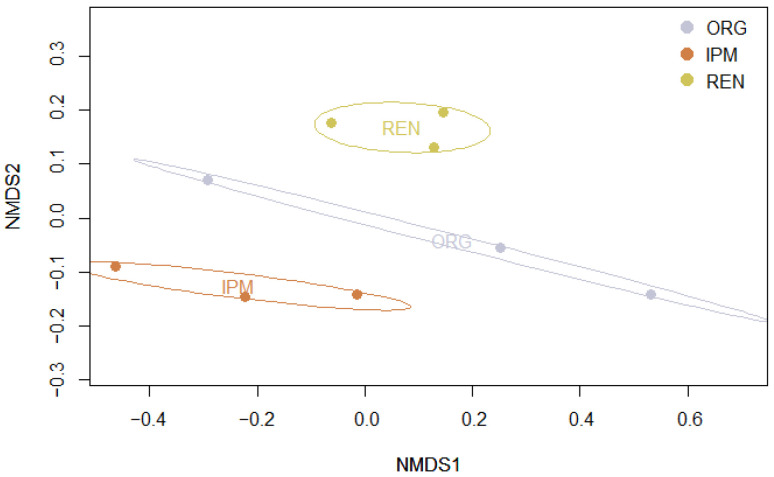
Non-metric multidimensional scaling (NMDS) of beetle communities based on Bray–Curtis dissimilarities. Points represent sites, coloured by management type (ORG, IPM, REN). Ellipses indicate 95% confidence intervals for each management type.

**Figure 9 insects-16-01148-f009:**
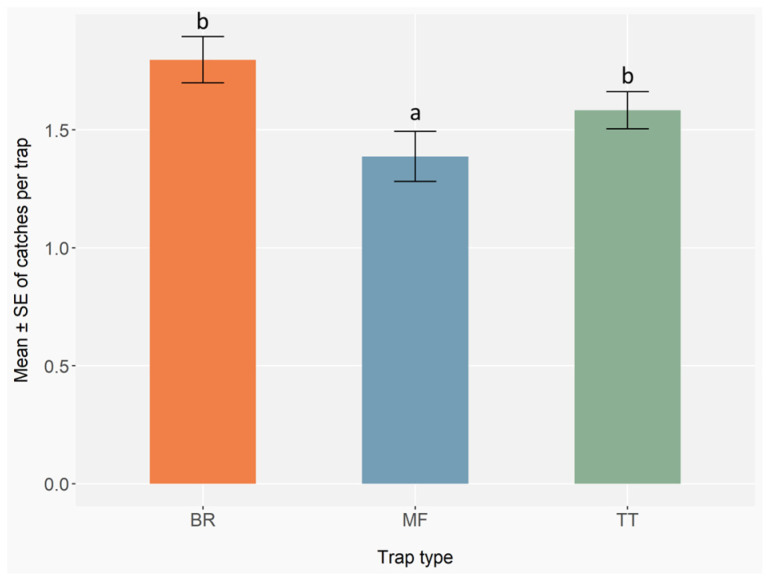
Mean number ± standard error (error bars) of ambrosia and bark beetles caught per trap type. Trap types include MF (multi-funnel, commercially available), BR (modified Brocap^®^) and TT (Tornado Tube^®^), both homemade models. Monitoring was conducted with 1 trap per type per block, 5 blocks per site, 9 sites, and 17 biweekly checks in 2023, and 1 trap per type per block, 3 blocks per site, 9 sites, and 17 checks in 2024. Different letters indicate statistically significant differences between trap types based on GLMER followed by Benjamini–Hochberg Post hoc comparisons (*p* < 0.05).

**Figure 10 insects-16-01148-f010:**
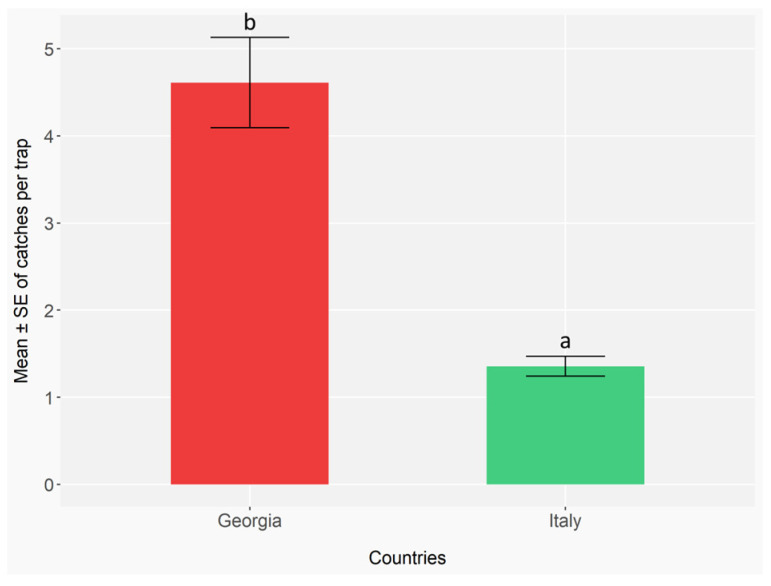
Mean number ± standard error (error bars) of ambrosia and bark beetles captured per trap, aggregated across all species, in the two countries (Italy and Georgia). In Italy, monitoring was conducted with 5 traps (TT) per site, 3 sites (IPM), and 17 biweekly checks in 2023, and 3 traps (TT) per site, 3 sites (IPM), and 17 checks in 2024. In Georgia, 3 traps (TT) per site, 5 sites (IPM), and 11 checks in 2023, and 3 traps (TT) per site, 4 sites (IPM), and 15 checks in 2024. Different letters indicate statistically significant differences between countries based on the Benjamini–Hochberg test (*p* < 0.05).

**Figure 11 insects-16-01148-f011:**
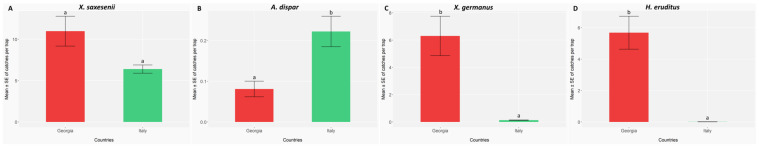
Mean number ± standard error (error bars) of ambrosia or bark beetle species caught per trap in Italy and Republic of Georgia: (**A**) *Xyleborinus saxesenii*; (**B**) *Anisandrus dispar*; (**C**) *Xylosandrus germanus*; (**D**) *Hypothenemus eruditus*. Different letters indicate statistically significant differences between countries based on GLMER followed by Benjamini–Hochberg Post hoc comparisons (*p* < 0.05).

**Table 1 insects-16-01148-t001:** Location, geographical coordinates, phytosanitary management, and year of observations of selected sites.

Site Code	Location	Geographical Coordinates	Phytosanitary Management	Year of Observation
ORG_1	Ischia di Castro (VT)—Italy	42°32′47.5″ N, 11°47′36.1″ E	Organic	2023–2024
ORG_2	Sutri (VT)—Italy	42°13′26.8″ N, 12°15′43.9″ E	Organic	2023
ORG_3	Bassano Romano (VT)—Italy	42°13′59.8″ N, 12°10′46.4″ E	Organic	2023–2024
ORG_4	Vetralla (VT)—Italy	42°17′52.8″ N, 12°05′58.4″ E	Organic	2024
IPM_1	Capranica (VT)—Italy	42°17′17.7″ N, 12°07′48.1″ E	IPM	2023–2024
IPM_2	Capranica (VT)—Italy	42°17′41.6″ N, 12°07′25.3″ E	IPM	2023–2024
IPM_3	Sutri (VT)—Italy	42°15′56.7″ N, 12°13′44.5″ E	IPM	2023–2024
REN_1	Calcata vecchia (VT)—Italy	42°12′34.2′′ N, 12°25′07.2″ E	Renaturalised	2023–2024
REN_2	Calcata vecchia (VT)—Italy	42°13′13.3″ N, 12°25′12.9″ E	Renaturalised	2023–2024
REN_3	Calcata vecchia (VT)—Italy	42°13′26.8″ N, 12°25′11.5″ E	Renaturalised	2023–2024
GEO_1	Jumi (Zugdidi)—Georgia	42°27′34.0″ N, 41°51′31.0″ E	IPM	2023–2024
GEO_2	Lia (Tsalenjikha)—Georgia	42°39′17.0″ N, 42°01′19.0″ E	IPM	2024
GEO_3	Kirovi (Zugdidi)—Georgia	42°25′03.0″ N, 41°38′55.0″ E	IPM	2023–2024
GEO_4	Zubi (Khobi)—Georgia	42°26′29.0″ N, 42°00′38.0″ E	IPM	2023
GEO_5	Sakhocholavo (Khobi)—Georgia	42°19′01.0″ N, 41°57′18.0″ E	IPM	2023–2024
GEO_6	Tsalenjikha (Tsalenjikha)—Georgia	42°34′32.0″ N, 42°01′49.0″ E	IPM	2023

## Data Availability

The data presented in this study are openly available in GitHub at https://github.com/eleonoracresta/BeetleCommunitiesHazelnutOrchards.git (accessed on 25 September 2025).

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
