# Peer review of "Bark and Ambrosia Beetle Communities in Hazelnut Orchards of Italy and Georgia"

_insects, 2025, doi:10.3390/insects16111148_

Round 1

Reviewer 1 Report

Comments and Suggestions for Authors

Review is attached.

Author Response

Author's Reply to the Review Report (Reviewer 1)

Dear Reviewer 1,

Thank you for your time dedicated to revise our manuscript, as well as for the suggestions and comments provided during the revision. We have sincerely appreciated all the indications, and we did our best, during the revision, to address all the comments.

A point-by-point answer follows below this document, and we renew our availability for any further change or request, if needed.

Thank you again for your time,

Sincerely

Nicolò Di Sora, on behalf of the coauthors.

  • L23 – deleted “small wood boring insects known as”. I think everyone knows bark beetles and what they look like.

Thank you for pointing out this issue. We agree with this comment; the sentence has been revised accordingly.

  • L25-27 - This statement is not based on truth, a lot of time and articles have been devoted to bark beetles and other pests on hazel. For example, Bucini et al. 2005 (Bio-ethology of Anisandrus dispar and Its Possible Involvement in Dieback (Moria) Diseases of Hazelnut (Corylus avellana L.) Plants in Central Italy) or AliNiazee 1997 (Integrated pest management of hazelnut pests: A worldwide perspective).

Thank you for this comment and for pointing out relevant studies on hazelnut pests. We have revised the sentence to better reflect the existing literature and to clarify that our study focuses on the broader beetle community and its relationship with different farming practices.

  • L47-48 - This sentence contradicts the sentence in L374-375 in the Discussion and at the same time with the Results. It needs to be edited.

Thank you for this observation. We would like to clarify that in lines 47–48, the number of captures reported for Italy refers to the total captures obtained from all sites (representing the three different management systems) and using all three types of traps. However, for an easier comparison with the data from Georgia (Section 3.4), we considered only the three IPM sites and captures obtained with TT traps. This difference in data aggregation explains the apparent discrepancy between the two sentences. To avoid any potential misunderstanding, we have removed the corresponding sentence from the Abstract.

  • L76-77 – part “currently considered transient, under eradication”. Reformulate it in a different form, because no one will ever eradicate the invasive bark beetles mentioned in Europe.

Thank you for this observation. We agree that the term “eradication” may be misleading in this context, as complete elimination of invasive bark beetles in Europe is unrealistic (and actually unfeasible). The sentence has therefore been revised to more accurately reflect the current phytosanitary status of these species.

  • Chapter 2.2 - It is not stated who determined the bark beetles, to correct it.

Thank you for this comment. The specialists who identified the specimens of bark and ambrosia beetles have now been cited in the Materials and Methods section.

  • L483 - The link to Github is not working, I recommend putting the data in the article as Supplementary material.

Thank you for reporting this issue. The GitHub link has been updated and is now fully working in the revised version of the manuscript.

  • Chapter Discussion: Georgia is an unexplored country in terms of bark beetles. In the Palaearctic catalogue of weevils, only Xylosandrus germanus is listed for Georgia, the other species are not listed there, but it is certain that they occur there. This discrepancy must be mentioned in the discussion.

Thank you for this comment. We agree that Georgia is relatively unexplored in terms of bark beetles. Indeed, in the Cooperative Catalogue of Palaearctic Coleoptera, Curculionoidea, 2nd edition, only Xylosandrus germanus is reported for Georgia. However, other species have been documented in the literature (e.g., https://doi.org/10.3390/f11030342, 2020; DOI: 10.3897/caucasiana.1.e85239). We have now added a statement in the Discussion to clarify this discrepancy.

  • It is also necessary to mention in the discussion the capture of Hypothenemus hampei in red traps, which this bark beetle prefers. What was the capture of eruditus in a red (BR) trap like?

Thank you for this suggestion. We have included some references to the known attraction of Hypothenemus hampei to red traps into the Discussion. However, we decided not to specifically mention H. eruditus, as our analysis did not show any significant differences between trap types for this species, probably due to its low number of captures.

Reviewer 2 Report

Comments and Suggestions for Authors

The manuscript addresses an applied and meaningful question for hazelnut IPM: how management regime (IPM/Organic/Renaturalised) and trap design (TT/BR/MF) influence bark and ambrosia beetle communities across two seasons and two countries (Italy, Georgia). The field effort is considerable and the topic is timely. However, several methodological issues in design and analysis weaken the current conclusions. I suggest Major Revision before the paper can be considered further.

Major comments

Sampling effort and time structure

Different deployment windows and different numbers of traps per year/site make “total catch” not directly comparable.

Please re-fit models at the check × trap level (15-day interval), with offset = log(trap-days) (or analyze catch per trap-day). This is essential to avoid biased effect sizes across traps, sites, years, and countries.

Site–year hierarchy and pseudo-replication

One organic site changed between years; thus site effects are partially confounded with year effects.

In GLMM, include site (orchard) and year as random factors (consider random slopes where appropriate). Blocks should be clearly nested. This will prevent attributing site/year variability to management.

Landscape and stand covariates

Some orchards are adjacent to natural forest; the “renaturalised” sites are within a protected area. Such differences (forest proximity, canopy closure, deadwood, tree vigor, irrigation) can strongly affect scolytines.

If these variables cannot be quantified now, please state this limitation clearly; if possible, include them as covariates.

Trap color vs. geometry confounding

BR (red, panel/funnel) and MF (black, multi-funnel) differ in both color and structure. The current design cannot separate color effects from geometry/airflow.

Please soften the conclusion about “color effect,” or add a small orthogonal test (color × geometry) as supplementary work.

Cross-country comparability

Only TT traps were used in Georgia, with different spacing and deployment windows from Italy.

Clarify the implications for Italy–Georgia comparisons and control them statistically (again, effort standardization is important).

Community-level analysis

Beyond totals and a few species, community patterns are not well demonstrated.

Please add α-diversity (Shannon/Simpson), β-diversity (Bray–Curtis) with PERMANOVA and NMDS/PCoA, and an indicator species analysis to support claims that “management shapes communities.”

Model diagnostics and multiplicity

Report checks for overdispersion and zero-inflation; provide residual diagnostics.

Present effect sizes with 95% CIs, not only letters from post-hoc tests.

Consider FDR (BH) alongside Bonferroni to show robustness.

Ethanol lure release and microclimate

A fixed 5 mL 96% ethanol renewed every ~15 days may evaporate at different rates across climates.

Please standardize with known-rate lures or acknowledge this as a limitation influencing comparability.

Taxonomic nomenclature

Unify the usage for species with synonyms (e.g., Anisandrus dispar vs Xyleborus dispar) and briefly explain the adopted nomenclature.

Strength of causal language

The result “Organic lowest; IPM/Renaturalised highest” is interesting, but given observational design and landscape differences, causal wording should be moderated. Emphasize association, not causation.

Minor comments

Abstract: State clearly that the cross-country comparison uses TT only in Georgia; avoid strong wording on color effects.

Figures: In captions, report n (traps/sites/checks). Provide a seasonal time series (per site or in supplement). Add full model tables.

Methods: Clarify the nesting of block/site/year/trapID, whether traps/blocks were reused between years, and exact distances to forest edges if available.

Data & code: In addition to GitHub, please archive a frozen copy on Zenodo (DOI) and cite the version/date for reproducibility.

Style: Keep English spelling consistent and define all abbreviations at first use (IPM, ORG, REN, TT, BR, MF, GLMM, etc.).

Actionable checklist (for revision)

Re-analyze counts at check × trap level with offset = log(trap-days); random effects: site, block (nested), trapID, year; evaluate zero-inflation where needed.

Add community metrics (α/β-diversity, PERMANOVA, NMDS/PCoA) and an indicator species analysis.

Author Response

Author's Reply to the Review Report (Reviewer 2)

Dear Reviewer 2,

Thank you for your time dedicated to revise our manuscript, as well as for the helpful comments and suggestions provided within your review. We have sincerely appreciated all the indications, and we did our best, during this round of revision, to carefully address all the points highlighted.

A point-by-point answer follows below in this document, and we renew our availability for any further change or request, if needed.

Thank you again,

Nicolò Di Sora, on behalf of the authors

Major comments

Sampling effort and time structure

  • Different deployment windows and different numbers of traps per year/site make “total catch” not directly comparable. Please re-fit models at the check × trap level (15-day interval), with offset = log(trap-days) (or analyze catch per trap-day). This is essential to avoid biased effect sizes across traps, sites, years, and countries.

Thank you for this comment. We clarify that the analysis was not based on total catches aggregated over the entire trapping period. The dataset included repeated measurements at each trap and date (15-day intervals), and the model was fitted using the number of individuals caught per trap and sampling event as the response variable. Therefore, differences in sampling effort were already accounted for in the analysis. We have rephrased the sentence in the Materials and Methods section to avoid misunderstanding.

In addition, we have included, in the Appendix, graphs showing the mean (± SE) number of individuals caught per trap for each management type and species across all sampling dates. These visualisations complement the statistical analyses by illustrating temporal trends and differences among management systems.

Site–year hierarchy and pseudo-replication

  • One organic site changed between years; thus site effects are partially confounded with year effects. In GLMM, include site (orchard) and year as random factors (consider random slopes where appropriate). Blocks should be clearly nested. This will prevent attributing site/year variability to management.

Thank you for this comment. We agree that site and year effects could be partially confounded, particularly because one organic orchard was replaced between years. To address this potential issue, we have updated all GLMMs to include both site and year as random effects, with blocks nested within sites (1 | Site/Block). This structure accounts the hierarchical design and prevents pseudo-replication, ensuring that variability attributable to sites or years is not incorrectly assigned to management effects.

Landscape and stand covariates

  • Some orchards are adjacent to natural forest; the “renaturalised” sites are within a protected area. Such differences (forest proximity, canopy closure, deadwood, tree vigor, irrigation) can strongly affect scolytines. If these variables cannot be quantified now, please state this limitation clearly; if possible, include them as covariates.

Thank you for this observation. We acknowledge that factors such as forest proximity, canopy closure, deadwood availability, tree vigour, and irrigation can influence bark beetle communities. Unfortunately, these variables were not quantified during our sampling and are therefore not included in the current models. We have added a statement in the Discussion (L417-422) to clearly acknowledge this limitation and to highlight that future studies could incorporate these covariates to further refine our understanding of management effects on scolytinae communities.

Trap color vs. geometry confounding

  • BR (red, panel/funnel) and MF (black, multi-funnel) differ in both color and structure. The current design cannot separate color effects from geometry/airflow. Please soften the conclusion about “color effect,” or add a small orthogonal test (color × geometry) as supplementary work.

Thank you for this comment. We acknowledge that BR (red, panel/funnel) and MF (black, multiple funnel) traps differ in both colour and structure and that, therefore, the differences observed in beetle capture could result from a combination of trap colour and trap geometry/airflow. We have therefore clarified, in the discussion, that the effect cannot be attributed to colour, only. Further studies would be needed to separate these factors, which have not been possible to carry out at this time.

Cross-country comparability

  • Only TT traps were used in Georgia, with different spacing and deployment windows from Italy. Clarify the implications for Italy–Georgia comparisons and control them statistically (again, effort standardization is important).

Thank you for this comment. The analyses were already carried out as suggested, standardising effort across traps and sites (L339-341). However, if we have misunderstood the point, we are happy to do further adjustments. For the Italy-Georgia comparison, only Italian sites under the same IPM management as in Georgia were considered, and only captures from TT traps, which are common to both countries, were included.

Community-level analysis

  • Beyond totals and a few species, community patterns are not well demonstrated. Please add α-diversity (Shannon/Simpson), β-diversity (Bray–Curtis) with PERMANOVA and NMDS/PCoA, and an indicator species analysis to support claims that “management shapes communities.”

Thank you for this suggestion. We have now included a full community-level analysis. Specifically, α-diversity (Shannon and Simpson indices) was calculated and compared among management types using Kruskal–Wallis tests. β-diversity was assessed with Bray–Curtis dissimilarity and tested using PERMANOVA. Community structure was visualised using NMDS. Finally, indicator species analysis (IndVal.g, indicspecies package) identified species significantly associated with each management type. These analyses consistently show that orchard management significantly influences beetle community composition.

Model diagnostics and multiplicity

  • Report checks for overdispersion and zero-inflation; provide residual diagnostics. Present effect sizes with 95% CIs, not only letters from post-hoc tests. Consider FDR (BH) alongside Bonferroni to show robustness.

Thank you for this suggestion. We have now included full model diagnostic checks in the text for overdispersion and zero-inflation, as well as residual diagnostics and 95% CIs. In addition, we have reported the complete tables of the tested models and post-hoc tests, and we have applied the Benjamini–Hochberg (BH) correction, as suggested.

Ethanol lure release and microclimate

  • A fixed 5 mL 96% ethanol renewed every ~15 days may evaporate at different rates across climates. Please standardize with known-rate lures or acknowledge this as a limitation influencing comparability.

Thank you for this comment. We agree that ethanol evaporation rates may vary across sites depending on local climate conditions (temperature, humidity, and wind exposure), potentially influencing trap attractiveness. Since a fixed 5 mL of 96% ethanol was used and replaced every ~15 days, we cannot exclude differences in evaporation rate among sites. We have added this as a methodological limitation in the discussion.

Taxonomic nomenclature

  • Unify the usage for species with synonyms (e.g., Anisandrus dispar vs Xyleborus dispar) and briefly explain the adopted nomenclature.

Thank you for this comment. The nomenclature has been standardised throughout the manuscript, adopting the currently accepted names.

Strength of causal language

  • The result “Organic lowest; IPM/Renaturalised highest” is interesting, but given observational design and landscape differences, causal wording should be moderated. Emphasize association, not causation.

Thank you for this comment. We have revised the discussion to moderate causal language and now describe the observed patterns as associations consistent with the observational nature of the study.

Minor comments

  • Abstract: State clearly that the cross-country comparison uses TT only in Georgia; avoid strong wording on color effects.

Thank you for this suggestion. We have revised the Abstract to clarify the trap types used in each country. In Italy, three trap designs were employed (handmade red funnel [BR], bottle [TT], and commercial multi-funnel [MF]), whereas in Georgia only TT traps were used.

  • Figures: In captions, report n (traps/sites/checks). Provide a seasonal time series (per site or in supplement). Add full model tables.

Thank you for this suggestion. We have revised and updated all figure captions to include the number of traps, blocks, sites, and checks per year. We have produced seasonal time series graphs showing the temporal trend of ambrosia and bark beetle captures per trap in hazelnut orchards under the three management systems (ORG, IPM, and REN). These graphs are included in the Appendix. The complete tables for all models have also been included in the Appendix.

  • Methods: Clarify the nesting of block/site/year/trapID, whether traps/blocks were reused between years, and exact distances to forest edges if available.

Thank you for this suggestion. We have added a sentence in the Materials and Methods section clarifying that in both Italy and Georgia, randomised blocks were nested within sites, and the same sites and blocks were monitored in both 2023 and 2024. Each block was placed at least 20 meters from the edge of the hazelnut orchard, as reported in section 2.2 (line 205).

  • Data & code: In addition to GitHub, please archive a frozen copy on Zenodo (DOI) and cite the version/date for reproducibility.

Thank you for this suggestion. At this stage, we prefer to provide the code exclusively via GitHub. The repository contains all scripts necessary to reproduce the analyses.

  • Style: Keep English spelling consistent and define all abbreviations at first use (IPM, ORG, REN, TT, BR, MF, GLMM, etc.).

Thank you for this comment. The manuscript text has been carefully checked for consistent English spelling, and all abbreviations (IPM, ORG, REN, TT, BR, MF, GLMM, etc.) have been defined at first use.

Actionable checklist (for revision)

Re-analyze counts at check × trap level with offset = log(trap-days); random effects: site, block (nested), trapID, year; evaluate zero-inflation where needed.

Add community metrics (α/β-diversity, PERMANOVA, NMDS/PCoA) and an indicator species analysis.

Thank you for this summary. Please refer to the previous answers for a complete tracking of the changes carried out during the revision.

Round 2

Reviewer 2 Report

Comments and Suggestions for Authors

This manuscript investigates scolytine communities in hazelnut under different management regimes (IPM, organic, renaturalised), compares three ethanol-baited trap designs, and provides a cross-country comparison (Italy vs Georgia). The objectives are clear and fit the journal scope; the experimental design is generally sound, and the statistical workflow (NB-GLMM with random effects, BH correction) is appropriate. Key findings—Xyleborinus saxesenii dominance, lower catches in organic orchards, and better performance of simple handmade traps—are practical for monitoring and IPM programs.

Minor comments

  1. You used 5 mL of 96% ethanol renewed biweekly. Because release rates vary with temperature/humidity, please add (i) a note on typical temperature/RH by site/date (or a summary climatic index) and (ii) whether lure mass-loss was spot-checked. This will help readers interpret between-site differences and the Italy–Georgia contrast. Also, specify detergent concentration consistently across sections.
  2. Trap design: colour vs geometry: Because BR and TT differ from MF in both colour and architecture, please strengthen the discussion that the observed performance likely reflects combined effects. If possible, add a short recommendation for growers (cost per unit, ease of deployment, maintenance frequency, and a schematic) so the applied audience can adopt the method.

  3. Methods clarity. State trap height, inter-trap spacing, and block distances once in a consolidated paragraph (some appear in multiple sections) and add a simple deployment diagram in SI for reproducibility.

Author Response

Response to Reviewer 2

Reviewer 2: This manuscript investigates scolytine communities in hazelnut under different management regimes (IPM, organic, renaturalised), compares three ethanol-baited trap designs, and provides a cross-country comparison (Italy vs Georgia). The objectives are clear and fit the journal scope; the experimental design is generally sound, and the statistical workflow (NB-GLMM with random effects, BH correction) is appropriate. Key findings—Xyleborinus saxesenii dominance, lower catches in organic orchards, and better performance of simple handmade traps—are practical for monitoring and IPM programs.

Response: Dear Reviewer 2, we are sincerely grateful for the time dedicated to revise our manuscript and for the constructive comments and suggestions as well. We have sincerely appreciated the suggestion regarding data analysis and sampling scheme.

We carefully integrated your comments and suggestions during the revision, with the hope to have sufficiently improved the overall quality of the manuscript. A point-by-point answer follows below in this document, and we renew our availability for any further change or request, if needed.

Minor comments – Rev2

Reviewer 2

  1. You used 5 mL of 96% ethanol renewed biweekly. Because release rates vary with temperature/humidity, please add (i) a note on typical temperature/RH by site/date (or a summary climatic index) and (ii) whether lure mass-loss was spot-checked. This will help readers interpret between-site differences and the Italy–Georgia contrast. Also, specify detergent concentration consistently across sections.

Response: Thank you for the for this useful suggestion. Following the comment, a table summarising the typical monthly minimum and maximum temperatures for the main locations of the study sites has been added in the Appendix. Unfortunately, consistent relative humidity data were not available for all locations, and thus could not be included. Regarding lure release, additional specifications were added in the text to clarify that the ethanol volume was renewed biweekly, but lure mass-loss was not systematically spot-checked. Furthermore, the detergent concentration has been standardised and consistently reported across all relevant sections. We hope that this new version may encounter your approval and thank you again for the suggestion.

Reviewer 2

  1. Trap design: colour vs geometry: Because BR and TT differ from MF in both colour and architecture, please strengthen the discussion that the observed performance likely reflects combined effects. If possible, add a short recommendation for growers (cost per unit, ease of deployment, maintenance frequency, and a schematic) so the applied audience can adopt the method.

Thank you for the comment. The section in the discussions dedicated to this part has been improved and provided with more information, to make easier for the audience the consultation of the methodology.

Reviewer 2

  1. Methods clarity. State trap height, inter-trap spacing, and block distances once in a consolidated paragraph (some appear in multiple sections) and add a simple deployment diagram in SI for reproducibility.

Thank you for this comment. We have revised the Methods section to consolidate all information on trap deployment into a single, clear paragraph. We also improved Figure 3, which already illustrated the deployment scheme, by refining its design and adding additional spatial details to enhance clarity and reproducibility. We hope that these section may appear now more clear and more suitable for the clarity of content.
